# Zero shot Generalization of Vision-Based RL Without Data Augmentation

## Abstract

Generalizing vision-based reinforcement learning (RL) agents to novel environments remains a difficult and open challenge. Current trends are to collect large-scale datasets or use data augmentation techniques to prevent overfitting and improve downstream generalization. However, the computational and data collection costs increase exponentially with the number of task variations and can destabilize the already difficult task of training RL agents. In this work, we take inspiration from recent advances in computational neuroscience and propose a model, Associative Latent DisentAnglement (ALDA), that builds on standard off-policy RL towards zero-shot generalization. Specifically, we revisit the role of latent disentanglement in RL and show how combining it with a model of associative memory achieves zero-shot generalization on difficult task variations *without* relying on data augmentation. Finally, we formally show that data augmentation techniques are a form of weak disentanglement and discuss the implications of this insight.

## 1 Introduction

Training generalist agents that can adapt to novel environments and unseen task variations is a long-standing goal for vision-based RL. RL generalization benchmarks have focused on data augmentation to increase the amount of training data available to the agent while preventing model overfitting and increasing robustness to environment perturbations (Yarats et al., 2021a; Almuzairee et al., 2024; Hansen et al., 2021). This follows the current trend in the broader robot learning community of training large models at scale on massive datasets (Kim et al., 2024; Hansen et al., 2024; Team et al., 2024) with the hope that the model will generalize. However, a significant drawback of these approaches is, intuitively, that they require larger models, more training data, longer training times, and have greater training instability that must be dealt with care when training RL agents.

Yet when we examine biological agents, we find that humans and, indeed, many other primates are able to quickly adapt to task variations and environment perturbations DiCarlo et al. (2012); Friston (2010). While all aspects of biological intelligence that contribute to generalization have yet to be understood, there is some understanding in the recent neuroscience literature of aspects related to representation learning that we look to for inspiration. Many parts of the brain in human and non-human primates contain neurons that represent single factors of variation within the environment, such as grid cells (Hafting et al., 2005), object-vector cells (Høydal et al., 2019), and border cells (Solstad et al., 2008) that represent euclidean spaces, distance to objects, and distance to borders, respectively. Such *disentangled representations* have been theorized to facilitate compositional generalization (Higgins et al., 2018) and have been studied with curated datasets where the factors of variation are known (Higgins et al., 2017a; Whittington et al., 2023), and even within the context of RL (Higgins et al., 2017b). It is then to our surprise, with limited exceptions (Dunion et al., 2023; Sax et al., 2018), that disentangled representation learning has not garnered much attention within robot learning or RL more generally. One potential reason for this is that learning disentangled representations while simultaneously learning an RL policy is extremely difficult. Indeed, Higgins et al. (2017b) required a two-stage approach where the disentangled representation was learned first, followed by policy learning. In addition, Yarats et al. (2021b) found that using a $\beta$-VAE directly led to training instability and worse performance, instead opting to use a deterministic autoencoder with softer constraints. Finally, there is counter-evidence by Schott et al. (2022) to suggest that, while disentangling representations may *facilitate* generalization, it alone cannot achieve out-of-distribution (OOD) generalization.

Figure 1: Disentanglement + association. A disentangled representation is learned using the original training data (top). When encountering an OOD sample (bottom), individual latents can be compared and mapped back to known values (colored green). Latent dimensions that are more OOD (colored red) can be mapped back without affecting other latent dimensions.

We hypothesize that one of the potential key missing ingredients to OOD generalization is *associative memory* mechanisms that use prior experiences to help inform decision-making in light of new data in a *disentangled* latent space. Intuitively, if the representation of a stored memory and new observation are disentangled, then the projection of the new observation onto the stored memories becomes a factorized projection in which individual factors can be compared independently of other factors (Figure 1). Indeed, many of the aforementioned single-factor neurons exist in the entorhinal cortex in the hippocampus (Hafting et al., 2005), responsible for, amongst other things, memory recollection and association. Recent literature suggests that the hippocampus learns flexible representations of memories by decomposing sensory information into reusable components and has been implicated in other cognitive tasks such as planning, decision-making, and imagining novel scenarios (Behrens et al., 2018; Rubin et al., 2014). It would then seem that the disentanglement of high-dimensional data into a modular, reusable representation is simply the first step in a multi-step process that enables generalization in biological agents. Inspired by these two ingredients found in nature, we propose a new method, ALDA (Associative Latent DisentAnglement), that 1. learns a disentangled representation from the training data and 2. uses an associative memory model to recover data points in the original training distribution zero-shot given OOD data. We demonstrate how this approach enables zero-shot generalization on a common generalization benchmark for vision-based RL without using data augmentation techniques or techniques that remove distractor variables from the latent space. Finally, we provide a formal proof showing that data augmentation methods for vision-based RL create what we refer to as "weak" disentanglement, where the latent space is perhaps partitioned into two or more categories but not perfectly factorized into individual subcomponents. We conclude by discussing the implications of this insight and future directions of this line of research.

## 2 BACKGROUND

### 2.1 REINFORCEMENT LEARNING

We wish to learn a policy $\pi$ that maps states to optimal actions that maximize cumulative reward. The agent-environment interaction loop is typically formulated as a Markov Decision Process (MDP) $(\mathcal{S}, \mathcal{A}, \mathcal{R}, \mathcal{P}, \gamma)$, where $\mathcal{S}$ and $\mathcal{A}$ are the state and action spaces, $\mathcal{R}(s, a)$ is the reward function, $\mathcal{P}(s'|s, a)$ is the probabilistic transition function, and $\gamma$ is the discount factor. The policy $\pi$ learns a mapping of state to action with the objective of maximizing cumulative discounted return $G_t = \mathbb{E}\left[\sum_{t=0}^{T} \gamma^t R(s_t, a_t)\right]$. In vision-based RL, we do not assume access to the low dimensional state $s_t \in \mathcal{S}$. Instead, we must infer $s_t$ given high-dimensional image observations $o_t \in \mathcal{O}$, making the problem a partially observable MDP, or POMDP $(\mathcal{S}, \mathcal{A}, \mathcal{R}, \mathcal{P}, \mathcal{O}, \gamma)$, where $\mathcal{O}$ is the space of high-dimensional observations.

**Soft Actor-Critic** (Haarnoja et al., 2018) is an off-policy actor-critic algorithm that jointly trains a policy $\pi$ and state-action value function $Q$ using the maximum entropy framework. The policy optimizes the maximum entropy objective $\arg\max_\pi \sum_{t=1}^{T} \mathbb{E}_{(s_t, a_t) \sim \rho_{pi}}[r_t + \alpha \mathcal{H}(\pi(\cdot|s_t))]$. The optimal

$Q$ function $Q^*(s, a)$ is estimated using temporal difference learning (Sutton, 1988) by minimizing the soft Bellman residual:

$$J(Q) = \mathbb{E}_{(s_t, a_t, s_{t+1}) \sim \mathcal{D}} \left[ (Q(s_t, a_t) - r_t - \gamma y)^2 \right].$$

Here, $y$ is the soft $Q$-target, which is computed as $y = r(s_t, a_t) + \gamma(\min_{\theta_{1,2}} Q_{\theta_i'}(s_{t+1}, a_{t+1}) - \alpha \log \pi(\cdot | s_{t+1}))$. We can then describe the policy's objective as:

$$J(\pi) = \mathbb{E}_{s_t \sim \mathcal{D}} \left[ \min_{i=1,2} Q_{\theta_i}(s_t, a_t) - \alpha \log \pi(a_t | s_t) \right].$$

A replay buffer $\mathcal{D}$ is maintained that contains transition tuples $(s_t, a_t, s_{t+1})$ collected from prior interactions of a potentially different behavior policy. Since the off-policy RL formulation does not require transitions to be from the current behavior policy, we can reuse prior experience to update the policy and the $Q$-function. We use SAC as a foundation for our method, and while we propose some architectural changes to improve the synergy between SAC and our method, the changes are generally applicable to most off-policy RL algorithms.

## 2.2 DISENTANGLED REPRESENTATION LEARNING.

**Nonlinear ICA**: The disentanglement problem is sometimes formulated in the literature (Hsu et al., 2023) through nonlinear independent component analysis (ICA) due to their conceptual similarity. We follow suit since the notation will be useful in later sections. Suppose there are $n_s$ nonlinear *independent* variables $s_1, ..., s_{n_s}$ that are the sources of variation of the images in the data distribution. A data-generating model maps sources to images:

$$p(\mathbf{s}) = \prod_{i=1}^{n_s} p(s_i), \mathbf{o} = g(\mathbf{s}) \tag{1}$$

where $g : \mathcal{S} \to \mathcal{O}$ is the non-linear data generating function. The nonlinear ICA problem is to recover the underlying sources given samples from this model. Similarly, the goal of latent disentanglement is to learn a latent representation $\mathbf{z}$ such that every variable $z_1, ..., z_{n_s} \in \mathbf{z}$ corresponds to a distinct source $s_1, ..., s_{n_s}$. Unfortunately, nonlinear ICA is *nonidentifiable* – that is, there are many decompositions of the data into sets of independent latents that fit the dataset, and so recovering the true sources reliably is impossible. Thus, the field of disentangled representation learning has focused more on empirical results and evaluation metrics on toy datasets where the true sources of variation are known. Given a dataset of paired source-data samples $(\mathbf{s}, \mathbf{o} = g(\mathbf{s}))$, the goal is to learn an encoder $f : \mathcal{O} \to \mathcal{Z}$ and a decoder $\hat{g} : \mathcal{Z} \to \mathcal{O}$ such that the disentanglement evaluation metrics are high while also maintaining acceptable reconstructions of the data. Disentanglement models are typically constructed as (variational) autoencoders (Whittington et al., 2023; Hsu et al., 2023; Higgins et al., 2017a) and are rarely applied outside of toy datasets.

## 2.3 GENERALIZATION IN VISION-BASED RL

Image augmentation methods have shown success and have become the go-to method for generalizing vision-based RL algorithms such as Soft Actor-Critic (SAC) (Haarnoja et al., 2018) and TD3 (Fujimoto et al., 2018), generally using augmentations such as random crops, random distortions, and random image overlays to simulate distracting backgrounds. Methods such as DrQ (Yarats et al., 2021a), SADA (Almuzairee et al., 2024), and SVEA (Hansen et al., 2021) regularize the $Q$ function by providing the original and augmented images as inputs into the critic. In many cases, however, the image augmentations can put the training data within the support of the distributions of the evaluation environments. For example, the "random convolution" image augmentation changes the color of the agent and/or background, and the policy is evaluated on an environment where the color of the agent is randomized. This brings into question whether these methods are truly capable of zero-shot *extrapolative* generalization when the training data is made to be sufficiently similar to the test data.

Beyond image augmentation techniques are methods that perform self-supervision using auxiliary objectives. Note that image augmentations for RL are also sometimes referred to as self-supervised

objectives, however we wish to make the distinction between methods that leverage data augmentation and those that don't. DARLA Higgins et al. (2017b), to the best of our knowledge, is the only prior method that learns a disentangled representation of the image inputs using a highly regularized $\beta$-VAE (Higgins et al., 2017a) for zero-shot generalization in RL. DARLA's approach is two-stage, where an initial dataset is collected by sampling random actions to first learn a disentangled latent representation, and then a policy is trained on this representation to maximize future return. However, a significant shortcoming is that random actions may not cover the full state distribution of the agent for more complicated tasks, whereas our method jointly learns the disentangled representation and the policy. SAC+AE (Yarats et al., 2021b) trains a decoder to reconstruct the images, resulting in a rich latent space that improves performance and sample efficiency on many vision-based tasks. Interestingly, SAC+AE mentions $\beta$-VAE's used by DARLA and proposes using a deterministic variant with similar constraints that shows some zero-shot generalization capability, but the authors make no mention of disentanglement and instead conclude that the key ingredient was adding a reconstruction loss as an auxiliary objective.

Another promising approach to generalization is learning a task-centric or object-centric representation using auxiliary objectives. Yamada et al. (2022) learn a task-centric representation by using expected discounted returns as labels, with the auxiliary task being to minimize the error between the predicted and true return values using the learned representation. Ferraro et al. (2023); Pore et al. (2024) use segmentation masks to learn object-centric representations that are robust to background distractors. One drawback of these approaches is that the latent representation overfits to the task by excluding all other information not relevant to the immediate task, usually citing that irrelevant information in the latent space hinders generalization performance. However, adapting to a new task that involves information that was previously considered irrelevant becomes a challenge for these methods. We hypothesize that the issue is not having "irrelevant" information in the latent space but rather that the latent variables are entangled without strong priors for disentanglement. A disentangled representation then paves the way for association, whereby individual dimensions of latent vectors from OOD images can be independently zero-shot mapped back to known values of those latent variables learned from the training data.

## 2.4 Associative Memory

An associative memory (AM) network stores a set of patterns with the intent to retrieve the most similar stored pattern given an input. The best-known form is a Hopfield network, originally proposed in Hopfield (1982), which was inspired by how the brain is capable of recalling entire memories given partial or corrupted input (e.g., recalling a food item given a particular smell). Classical Hopfield networks could only store and recall binarized memories, whereas modern (dense) Hopfield networks (Krotov & Hopfield, 2016) can work with continuous representations and are trainable as differentiable layers within existing Deep Learning frameworks (Ramsauer et al., 2021). The memory retrieval dynamics are typically formulated as a function of energy minimization. Let $\xi \in \mathbb{R}^d$ be the input query pattern, and $\mathbf{X} := [x_1, ... x_M] \in \mathbb{R}^{d \times M}$ be memory patterns. In AM models, memories are stored on the local minima of the energy landscape, where the goal is to retrieve the closest stored pattern to $\xi$ by minimizing energy. Modern Hopfield networks assume the following general form for the energy function:

$$E = -\sum_{i=1}^{M} F(x_i^T \xi).$$

In particular, by setting $F = -lse(\beta, \mathbf{X}^T \xi) + \frac{1}{2} \xi^T \xi$ ($lse$ = log-sum-exponent), the retrieval dynamics becomes $\xi^{new} = \mathbf{X} \text{softmax}(\beta \mathbf{X}^T \xi)$, which is the attention mechanism (Vaswani, 2017) and the backbone of modern Hopfield networks. Follow-up works such as Bietti et al. (2023) show a tight connection between the learning dynamics of Transformers and models of associative memory.

# 3 On the Relationship Between Disentanglement and Data Augmentation

We begin by motivating the case for learning a disentangled representation for RL agents by showing a connection between data augmentation and disentangled representation learning. Specifically, we formally prove that data augmentation is a *weak* disentanglement of the latent space. We define weak

disentanglement as a partial factorization of some but perhaps not all latent dimensions of the latent space i.e. $\exists z_i \in \mathbf{z}, s_j, s_k \in \mathbf{s} | cov(\hat{s_j}, \hat{s_k} | z_i) \neq 0$. Strong disentanglement, on the other hand, is a complete factorization where each latent dimension $z_k \in \mathbf{z}$ encodes for a unique source $s_i \in \mathbf{s}$ and is thus linearly independent of other latent dimensions, which is the goal of disentangled representation learning. The full proof is provided in A.1.

**Theorem 1**: Suppose we are given a $\mathbf{z} = f_\theta(g(\mathbf{s}))$, where some latent dimension $z_k \in \mathbf{z}$ approximates one or more sources. We will denote the approximations as $\hat{s_i}$. We can categorize the sources $\mathbf{s}$ into two categories, $D$ and $E$, which correspond to task-relevant and task-irrelevant sources, respectively. For any such $z_k$, if $Q^*(\mathbf{z}, a)$ is an *optimality invariant optimal Q-function* immune to distractor variables, then the following must be true of $\mathbf{z}$:

$$cov(\hat{s_i}, \hat{s_j} | z_k) = 0 \; \forall s_i \in D, s_j \in E, z_k \in \mathbf{z}. \tag{2}$$

Intuitively, if data augmentation enables learning a latent representation such that the $Q(\mathbf{z}, a)$, a function of $\mathbf{z}$, is immune to distractor variables, then any dimension of the latent space that encodes for task-relevant variables cannot also encode for task-irrelevant variables. Otherwise, distribution shifts involving the task-irrelevant variables would affect the $Q$ function and, thus, the performance of the agent. One of two conditions must be true: either $\mathbf{z}$ is partitioned, where some variables approximate only sources from $D$, and others only sources from $E$, or $\mathbf{z}$ contains no information about sources in $E$ altogether, both of which are a form of weak disentanglement.

We take a probabilistic perspective to see why this relationship is important. Suppose $s_{1...,k} \in D$ and $s_{1+k,...,n_s} \in E$. In order to learn a latent representation that contains no information about task-irrelevant sources $s_{1+k,...,n_s}$, data augmentation methods essentially estimate the marginal distribution over task-relevant sources:

$$p(s_1, ..., s_k) = \sum_{s_{k+1}} \times \sum_{s_{k+2}} \times ... \times \sum_{s_{n_s}} p(s_1, ..., s_k, s_{k+1}, ...s_{n_s}). \tag{3}$$

The implication of 3 is that we must collect data for *every* possible variation of the task-irrelevant sources, which may be prohibitively expensive for real-world applications. Instead, a model with strong priors that leads to disentanglement without data augmentation essentially achieves the same result without the additional costs. Although the latent representation of such a model may still contain task-irrelevant features, the policy can learn to simply ignore them or associate them with known values in the presence of OOD data, as is the case with ALDA. In addition, these task-irrelevant features may become relevant if the task changes (e.g., if the current task is for a manipulator to stack a blue cube, and the next task is to stack a red cube), and so it may, in fact, be important to keep them.

## 4 METHOD

**Experimental Setup**. We first describe the generalization benchmark and our evaluation criteria to provide additional context. We train on four challenging tasks from the DeepMind Control Suite (Tassa et al., 2018). To evaluate zero-shot generalization capability, we periodically evaluate model performance under challenging distribution shifts from the DMControl Generalization Benchmark (Hansen & Wang, 2021) and the Distracting Control Suite (Stone et al., 2021) throughout training. Specifically, we have two evaluation environments: **color hard**, which randomizes the color of the agent and background to extreme RGB values, and **distracting cs**, which applies camera shaking and plays a random video in the background from the DAVIS 2017 dataset (Pont-Tuset et al., 2017).

### 4.1 DISENTANGLEMENT

We now describe our framework for jointly learning a disentangled representation and performing association. For latent disentanglement, we choose to use QLAE Hsu et al. (2023), the current SOTA disentanglement method, which trains an encoder $f_\theta$ that maps to a continuous disentangled latent space, a discrete, parameterized latent model $l_\psi$, and a decoder $g_\phi$ that reconstructs the observation. Similar to VQ-VAE (van den Oord et al., 2017), QLAE uses a discrete codebook for the latent space, except that each dimension uses its own separate scalar codebook. Concretely, $Z$ is the set of latent

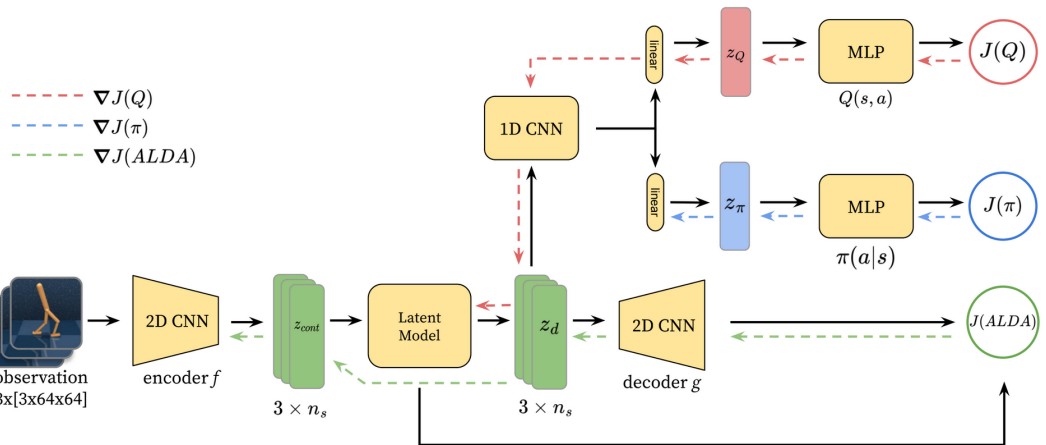

Figure 2: Diagram of our method SAC+ALDA. Trainable components are colored yellow. A strongly regularized autoencoder and the quantized latent space enable latent disentanglement. The latent model is also responsible for association when encountering OOD inputs.

codes defined by the Cartesian product of $n_z$ scalar codebooks $Z = V_1 \times ... \times V_{n_z}$ i.e., there are $n_z$ latent variables and $|V_j|$ discrete categories per variable. The continuous outputs of the encoder are the latent variables, each of which is quantized to the nearest scalar value in their respective codebooks.

$$z_{d_j} = \text{argmin}_{v_{jk} \in \mathbf{v}_j} |f_\theta(x)_j - v_{jk}|, j = 1, ..., n_z. \tag{4}$$

Since we cannot differentiate through $argmin$, as with VQ-VAE, the authors of QLAE use quantization and commitment losses and a straight-through gradient estimator (Bengio et al., 2013):

$$\mathcal{L}_{\text{quantize}} = ||\text{StopGradient} f_\theta(x)) - z_d||_2^2, \ \mathcal{L}_{\text{commit}} = ||f_\theta(x) - \text{StopGradient}(z_d)||_2^2. \tag{5}$$

The authors claim that while this is a failure mode for vector quantization, $Z$ is low-dimensional enough that, in practice, it does not meaningfully impact performance. While this may be true for standalone disentanglement benchmarks, we find that it causes training instability and performance degradation when jointly learning a policy for high-dimensional continuous control problems. We propose a solution in section 4.2 from the viewpoint of associative memory.

It is common practice in many vision-based RL algorithms to utilize framestacking to incorporate temporal information into the latent space. This means that the encoder accepts as input, and the decoder produces a stack of RGB images in $\mathbb{R}^{B \times Ck \times H \times W}$, where $k$ is the number of frames. However, latent disentanglement models have only been shown to work on datasets of singular images and struggle to disentangle sources of individual images when given stacks of images as inputs. Evidence of this is presented in the appendix, Section A.5. To resolve this issue, we fold $k$ into the batch dimension and encode/decode batches of single images in $\mathbb{R}^{Bk \times C \times H \times W}$, resulting in a batch size $Bk$ of disentangled latent vectors $z_d \in \mathbb{R}^{Bk \times n_{s_i}}$. To incorporate temporal information, we reshape the batch of latent vectors into $\mathbb{R}^{B \times kn_{s_i}}$ and feed it into a 1D convolutional neural network (CNN), producing our final latent vector $z \in \mathbb{R}^{B \times e}$. $z$ is used as the state representation for the actor and critic networks, while the decoder network for the disentanglement model only ever receives the disentangled representation $z_d$ as input.

## 4.2 ASSOCIATION

The naive approach to performing association would be to feed the quantized latent representation through a Hopfield network. However, upon closer inspection of QLAE's latent dynamics, we find that most of the components of a generic associative memory model are already present, i.e., QLAE is implicitly also a Hopfield network.

Figure 3 shows a comparison of using SAC with QLAE vs with BioAE Whittington et al. (2023), another disentanglement method that uses biologically inspired constraints and achieves comparable results on disentanglement benchmarks (Hsu et al., 2023). BioAE achieves strong initial performance on the two evaluation environments, but slowly degrades over the course of training. We suspect both models overfit to the training environment, but QLAE is capable of zero-shot mapping OOD latent variables to in-distribution values. To see the similarity, we first present the general framework described in Millidge et al. (2022) of a universal Hopfield

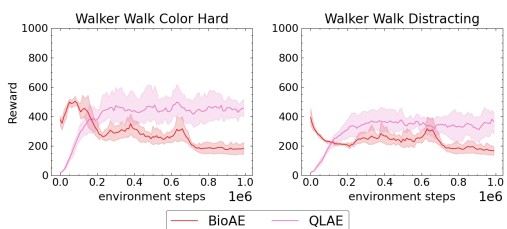

Figure 3: Ablation comparing SAC+QLAE to SAC+BioAE on the two distribution shift evaluation environments for the walker walk task.

network that all feedforward associative memory networks in the literature can be factorized as:

$$z = P \cdot sep(sim(X, \xi)). \tag{6}$$

$P$ is a projection, *sep* is a separation function, and *sim* is a similarity function between the stored memories $X$ and query $\xi$. While $P$ is originally described as a projection matrix, we extend the definition of $P$ to be any function that projects $X$ and $\xi$ into a shared embedding space. In this case, equation 4 can be interpreted as follows: $f_\theta$ is the projection function that projects high-dimensional images $\mathbf{o}$ into an embedding space shared by the scalar codebooks $Z$, which can be interpreted as predetermined memories. The closest memory is recovered by computing the $L_1$ distance, which serves as the similarity function, and *sep* is the *argmin* function. Through this view, we can rewrite the latent dynamics of QLAE in many ways, perhaps exchanging the similarity function for $L_2$ distance or dot product, changing the separation function, etc., as long as it follows the framework of 6. Since $Z$ is a product of scalar codebooks, $L_1$ distance remains an appropriate choice for the similarity function. Instead, we augment the latent dynamics with a Softmax separation function as follows:

$$z_{d_j} = \text{Softmax}(-\beta L_1(f_\theta(\mathbf{o})_j, \mathbf{v}_j)) \odot \mathbf{v}_j \tag{7}$$

where $\beta$ is a scalar temperature parameter. Equation 7 can be interpreted in two ways. From an associative memory perspective, attention-based Hopfield models apply Softmax to separate the local minima (stored memories) on the energy landscape, where $\beta$ controls the degree of separation, and so we've recovered the modern Hopfield memory retrieval dynamics. From a purely mathematical perspective, we have what resembles the Gumbel-Softmax categorical reparameterization Jang et al. (2017), although we do not perform any sampling in our method. This lends a novel view on attention-based Hopfield networks – models with a high-temperature parameter can be interpreted as classifiers over $|\mathbf{X}|$ classes, where $|\mathbf{X}|$ is the number of stored memories whose local minima are well separated on the energy landscape.

In the limit, as $\beta$ goes to infinity, we achieve maximum separation between memories and recover equation 4. In practice, we choose a large value for $\beta$ such that we retrieve one scalar from each codebook, as originally intended, although our method works well with smaller values of $\beta$ (see appendix Section A.4 for additional results). Since large $\beta$ values can cause downstream gradients to vanish, we find that keeping the commitment loss from equation 5 helps keep the outputs of the encoder close to the values of the latent model. However, we do not optimize the codebook towards the encoder outputs i.e., we omit $\mathcal{L}_{\text{quantize}}$. This can be interpreted as having a set of task-optimized memories that the encoder must learn to map to under the Hopfield interpretation. Our final objective for ALDA is as follows:

$$\begin{aligned} J(ALDA) &= \mathcal{L}_{\text{commit}} + \mathcal{L}_{\text{reconstruct}} \\ &= \mathbb{E}_{\mathbf{o}_t \sim \mathcal{D}}\Big[ ||\text{StopGradient}(f_\theta(\mathbf{o})) - [\text{Softmax}(-\beta L_1(f_\theta(o), V)) \odot V]||_2^2 \\ &\quad + \log g_\phi(\mathbf{o}_t|\mathbf{z}_d^t) + \lambda_\theta ||\theta||^2 + \lambda_\phi ||\phi||^2 \Big] \end{aligned} \tag{8}$$

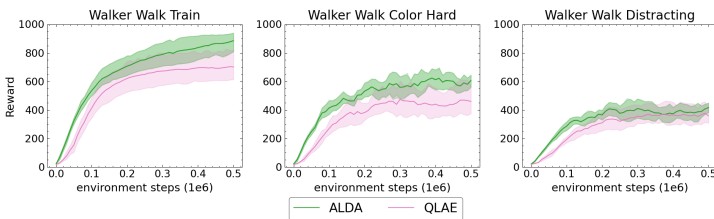

Figure 4: Ablation comparing ALDA and QLAE. Average of 5 seeds, shaded region represents a 95% confidence interval.

where the last two terms are weight-decay on the encoder and decoder parameters controlled by $\lambda_\theta$ and $\lambda_\phi$, respectively. Observations collected by the policy are stored in a replay buffer $\mathcal{D}$, from which batches are randomly sampled to train ALDA. We observe that this formulation considerably improves training and evaluation performance on the "color hard" environment and to a degree, on the DistractingCS environment, as shown in Figure 4.

The obvious question is, how do we set the dimensionality of $z_d$, which should 1:1 correspond to the number of sources $n_s$ if the number of sources is unknown? While there is no rigorous method to derive $|z_d|$ at this time, we empirically found that setting $|z_d|$ to within the ballpark of the size of the observation spaces for the proprioceptive, *state-based* versions of the tasks seemed to work well. $|z_d|$ is set to 12 for all reported tasks.

## 5 EXPERIMENTS

We compare against several baselines that together represent the full range of different learning paradigms in the literature that attempt to elicit zero-shot generalization. **DARLA** (Higgins et al., 2017b) is, to the best of our knowledge, the only other algorithm that attempts to learn a disentangled representation of the image distribution towards zero-shot generalization of vision-based RL. **SAC+AE** (Yarats et al., 2021b) uses a deterministic autoencoder with an auxiliary reconstruction objective and strong regularization that demonstrates decent zero-shot generalization capability. **RePo** (Zhu et al., 2023) is a model-based RL algorithm that learns a task-centric latent representation immune to background distractors. Finally, **SVEA** (Hansen et al., 2021) is an off-policy RL algorithm that improves training stability and performance of off-policy RL under data augmentation. As in their paper, we use the random overlay augmentation for SVEA, where images sampled from the Places (Zhou et al., 2017) dataset of 10 million images are overlayed during training. The training curves and evaluation on "color hard" and DistractingCS are presented in Figure 5.

Excluding SVEA, ALDA outperforms all baselines on both distribution shift environments. ALDA also maintains stability and high performance on the training environment, despite the disentanglement auxiliary objective and extremely strong weight decay ($\lambda_\theta, \lambda_\phi = 0.1$) on the encoder and decoder. We do not expect to outperform SVEA since it uses additional data sampled from a dataset of 1.8 million diverse real-world scenes, likely putting the training data within the support of the data distributions of the evaluation environments. Nevertheless, ALDA performs comparably and, in some cases, is equal to SVEA despite only seeing images from the original task. Performance degrades severely for all algorithms on the DistractingCS environment. We suspect that, in addition to the already difficult task of ignoring the background video, camera shaking affects the implicitly learned dynamics, and thus, additional finetuning may be unavoidable for this task. Still, ALDA performs better than all baselines excluding SVEA on Distracting CS as well, even matching the performance of SVEA on cartpole balance and finger spin.

The disentangled representation learning field primarily uses toy datasets where the ground truth sources of the data distribution are known. Therefore, all disentanglement metrics we are aware of require knowing the sources, making it difficult to quantitatively evaluate disentanglement performance on DMControl. In the absence of any quantitative disentanglement metrics, we opt to show empirical evidence of disentanglement in our model, presented in Figure 6. In this experiment, we encode an observation $\mathbf{o}$ into the disentangled latent representation $z_d = l_\psi(f_\theta(\mathbf{o}))$. We pick a latent variable $z_{d_i}$, traverse it while holding all other latent variables fixed, and decode the resulting latent vectors

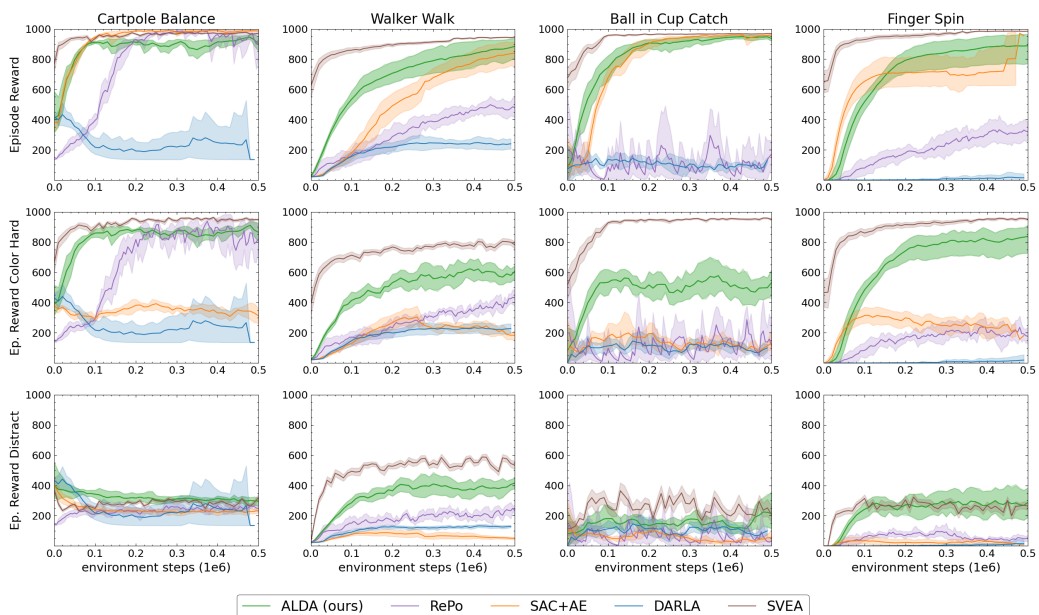

Figure 5: Top row: performance on the training environment. Middle and bottom rows are evaluation results on the "color hard" and DistractingCS evaluation environments, respectively. Average of 5 seeds, shaded region represents a 95% confidence interval.

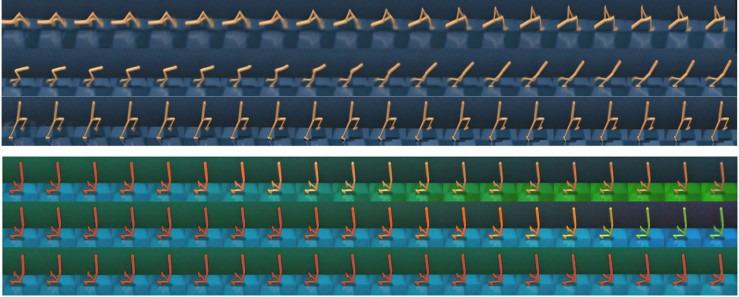

Figure 6: Visualization of different latent perturbations. **Top:** Traversal of select latents for the standard training environment. **Bottom:** Traversal of select latents when training directly on the color hard environment. Latents that encode for distractor variables (e.g., color) seemingly do not simultaneously encode for task-relevant variables (e.g., agent dynamics). Visualizations of all latent traversals can be found in Appendix section A.3.

into reconstructed observations $\mathbf{o}' = g_\phi(z_d')$. We find that each latent tends to learn information about a single aspect of the robot, for example, the orientation of the torso or the rotation of the left/right leg. We also find empirical evidence that ALDA does not discard task-irrelevant information, but rather encodes it separately from task-relevant latent variables when training ALDA directly on the color hard environment.

## 6 DISCUSSION

As stated previously, the disentanglement problem by extension of nonlinear ICA is underdetermined, so there are many ways the latent space may factorize, perhaps by representing the sky and background with one latent or by separating them into two latents, etc. Given that both the task and reconstruction gradients of the critic/decoder affect the latent model/encoder, an interesting scientific and philosophical implication is that the model is potentially biased towards a disentangled representation that is *useful*, although there is no way to quantitatively or qualitatively show such a

result at this time. Nevertheless, it remains an interesting line of further investigation from a scientific standpoint, and perhaps, philosophically, says something about whether the question "What is the ground truth factorized representation that best explains the data?" is even the right question to ask.

RL agents deployed in the real world must constantly adapt to changing environmental conditions. Much of the variance can be captured with a sufficiently large dataset. However, there remains a portion of the distribution containing every possible edge case and unaccounted-for variation, commonly referred to as "the long tail," that remains elusive because it is prohibitively expensive to account for every possible variation. Unfortunately, these uncaptured variations are frequent enough due to the ever-changing dynamical nature and complexity of the real world that deploying agents in the real world remains challenging. Therefore, it seems the case that data augmentation techniques, collecting massive datasets, and the like are not sufficient to develop generalist agents capable of adaptation the way humans and other animals are. That's not to say that data isn't important or a fundamental ingredient to training machine learning models. In fact, the method proposed in this paper scales with more data as with prior works that leverage data augmentation techniques. Instead, our proposition is that *if a data-driven model can generalize better with less data, then it will scale better with more data.*

In Section 3, we showed how data augmentation and disentangled representation learning aim to achieve the same result – a factorization of the latent space into separate components in order to improve downstream generalization performance. Given the additional computational and data collection costs and potential training instabilities that data augmentation methods may incur, it seems more fruitful to investigate models with inductive biases that elicit modular and generalizable representations without relying on data scaling laws. While presenting the model with sufficiently large and diverse datasets remains unquestionably important, we cannot rely solely upon the data in hopes that the model learns a good representation. As with any other inductive biases, such as using CNNs for vision tasks or transformers for NLP tasks, inductive biases that elicit modular representations *while leveraging data* are worth studying if we are to develop agents that can perform and adapt well in the real world.

We hope that the work presented here inspires future research into novel models and architectures to learn representations that enable the adaptability we see in our biological counterparts. We discuss some limitations of our method and promising directions for future research. One notable limitation is that our disentangled latent representation $z_d$ does not explicitly account for temporal information since it primarily estimates the sources that produce the image distribution. Instead, we must capture temporal information in the downstream 1D-CNN layer as shown in Figure 2. How to learn a disentangled representation that contains sources of both the image data and temporal information for decision-making tasks remains an open question. Another limitation is that, while we introduce a simple Hopfield model as a modification to QLAE, we do not take advantage of the more recent literature involving learnable attention-based or energy-based Hopfield networks (Ramsauer et al., 2021; Hoover et al., 2024). Stronger Hopfield models that synergize well with disentangled representations is another potentially fruitful research direction.

Given that we use a very compact disentangled latent space with strong empirical evidence that individual latents capture information about specific aspects of the agent, an interesting research direction is to investigate whether all or parts of the proprioceptive state representation can be recovered from image observations. We provide some preliminary evidence of this in the appendix (A.2). Beyond interpretability, such a model may yield better performance since state-based RL agents tend to perform better than vision-based agents. Finally, while our work was inspired by the role of the hippocampus in biological intelligence, the exact mechanisms of the machinery and how they interact with decision-making, planning, and imagination components of biological brains are by no means precisely modeled in this paper, nor are all of the computations the hippocampus may be performing fully understood. Future collaborative research between the machine learning and neuroscience fields into data-driven computational models of these mechanisms may yield even better-performing, adaptable agents.

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

# A   APPENDIX

## A.1   PROOF

**Preliminaries.** We reintroduce the nonlinear ICA problem formulation here for the reader's reference. There are $n_s$ nonlinear independent variables $\mathbf{s} = s_1, ..., s_{n_s}$ that are considered the "ground truth" sources of variation of the data distribution. We assume there exists a data-generating model that maps sources to images:

$$p(\mathbf{s}) = \prod_{i=1}^{n_s} p(s_i), \mathbf{o} = g(\mathbf{s})$$

where $g : \mathcal{S} \to \mathcal{O}$ is the non-linear data generating function. For the purpose of this proof, we restrict $\mathcal{S}$ and $\mathcal{O}$ to be the space of sources and images within our dataset. The goal of nonlinear ICA, and by extension, disentangled representation learning, is to recover the underlying sources given samples from this model. We claim that most, if not all, data augmentation techniques in $Q$-learning are a form of weak disentanglement, where either the method factorizes the latent space into task-relevant / task-irrelevant variables or removes task-irrelevant variables from the latent space entirely.

For a given task, we can split the sources into two categories: sources $s_{1...k}, k < n_s$ that are task-relevant, which we will call $D$, and sources $s_{k+1,...,n_s}$ that are not, whose category we will refer to as $E$. The encoder maps observations to a latent space $f : \mathcal{O} \to \mathcal{Z}$, and so $\mathbf{z}$ is a function of the sources $\mathbf{z} = f_\theta(g(s))$. We refer to $\hat{s}_i$ as an approximation to the true variable $s_i$ that exists in one or more dimensions of $\mathbf{z}$. We make no assumptions on whether the sources are entangled or disentangled in $\mathbf{z}$.

**Optimality Invariant Image Transformations** Described in Yarats et al. (2021a), data augmentation applied to $Q$-learning can be formulated using the following general framework. An optimality-invariant state transformation $h : \mathcal{O} \times \mathcal{T} \to \mathcal{O}$ is a mapping that preserves $Q$ values.

$$Q(f_\theta(\mathbf{o}), a) = Q(f_\theta(h(\mathbf{o}, v)), a) \forall \mathbf{o} \in \mathcal{O}, a \in \mathcal{A}, v \in \mathcal{T}. \tag{9}$$

$v$ are the parameters of $h(\cdot, \cdot)$ drawn from the set of all possible parameters $\mathcal{T}$. In other words, $\mathcal{T}$ defines the space of all possible data augmentations that should not affect the output of the $Q$-function.

**Proposition 1**: Let $\phi : S \to S$ be a function that perturbs sources $s_j \in E$. Then

$$h(\mathbf{o}, v) = g(\phi(\mathbf{s})). \tag{10}$$

This follows from the definition of $E$ in that any optimality-invariant transformation to the observation must have implicitly resulted from a perturbation of some task-irrelevant source $s_j \in E$.

**Proposition 2**: For any given $\mathbf{z} = f_\theta(g(\mathbf{s}))$ and any perturbation to a true source $s_j \in E, j \in [k+1, n_s]$ resulting in a new latent $\mathbf{z}' = f_\theta(g(\mathbf{s}'))$, the following must be true for an optimality invariant optimal $Q$-function:

$$Q^*(\mathbf{z}, a) = Q^*(\mathbf{z}', a). \tag{11}$$

We can rewrite $\mathbf{z}'$ as $\mathbf{z}' = f_\theta(g(\phi(\mathbf{s})))$ i.e. $\mathbf{o} = g(\phi(\mathbf{s}))$, and by Proposition 1, $g(\phi(\mathbf{s})) = h(\mathbf{o}, v)$. Essentially, an optimality invariant $Q$-function is immune to variations of task-irrelevant sources from the set $E$, since they correspond to optimality-invariant state transformations.

**Theorem 1**: For any $\mathbf{z} = f_\theta(g(\mathbf{s}))$, and for any dimension $z_k$ of $\mathbf{z}$, the following must be true for an optimality invariant $Q$-function:

$$cov(\hat{s}_i, \hat{s}_j | z_k) = 0 \ \forall s_i \in D, s_j \in E, i \in [1, k], j \in [k+1, n_s]. \tag{12}$$

To see why this must be the case, suppose that the covariance is nonzero and suppose that we perturb $s_j \in E$ to $s'_j$, giving us a new observation $\mathbf{o}' = g(\mathbf{s}')$. Since $s_j$ is task-irrelevant, $\mathbf{z}' = f_\theta(\mathbf{o}') = f_\theta(h(\mathbf{o}, v))$ for some $v \in \mathcal{T}$. If $\mathbf{z}'$ is a function of $h$, then by equations 9 and 11,

$Q^*(\mathbf{z}', a) = Q^*(\mathbf{z}, a)$. However, if $cov(\hat{s_i}, \hat{s_j}|z_k) \neq 0$ for some $z_k \in \mathbf{z}$, then $\hat{s_i}' \in \mathbf{z}' \neq \hat{s_i} \in \mathbf{z}$. Since $\hat{s_i} \in D$, $Q^*(\mathbf{z}', a) \neq Q^*(\mathbf{z}, a)$, which is a contradiction. Therefore, the conditional covariance between any $\hat{s_i} \in D$ and any $\hat{s_j} \in E$ for any given $z_k$ must be zero, which implies that the approximations of task-relevant and task-irrelevant sources in $\mathbf{z}$ are disentangled.

## A.2 LATENT TRAJECTORY VISUALIZATIONS

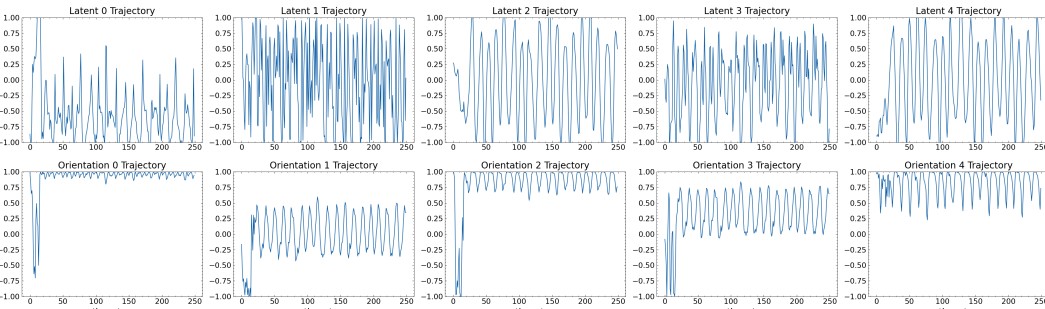

Figure 7: Visualizations of a few latent/state trajectories through time for the Walker agent. **Top**: Trajectories of several latent variables from the disentangled latent space. **Bottom**: Trajectories of several rigid body orientations.

Given the low dimensionality of the disentangled latent representations and the fact that they are disentangled, we hypothesize that their trajectories through time may correspond to trajectories of proprioceptive state variables such as rigid-body positions and orientations. We visualize the trajectories of individual latents through time for a single episode alongside the trajectories of several proprioceptive state variables in Figure 7. Unfortunately, the mappings of sources to the disentangled latent space likely do not correspond 1:1 with the proprioceptive state, given that the learned mappings are arbitrary and not unique. However, upon visual inspection, we find that latent trajectories through time exhibit oscillatory behavior patterns similar to that of rigid body orientations from the state representation. Recovering all or parts of the proprioceptive state representation via unsupervised learning from high-dimensional data is an interesting future research direction.

### A.3 Additional Latent Traversal Plots

Latent traversals for the other reported DMControl tasks are presented here. We also visualize the latent traversals of ALDA trained directly on the color hard environment. We find that the latent traversals for cartpole balance are more discontinuous than on other tasks. One reason for this might be the lack of balanced data and data diversity of the cartpole replay buffer. The (near) optimal policy is achieved quite early on, after which most images collected are of the cartpole upright and roughly in the same $x$-position. The lack of data diversity likely makes it difficult to learn a representation in which the latent traversals are more continuous / physically plausible. Interestingly, this phenomenon does not seem to affect performance on the "color hard" evaluation environment, although we suspect there are performance gains to be had on DistractingCS if the latent interpolations were smoother. We leave an investigation into the effects of data balancing and data diversity on downstream generalization performance as future work.

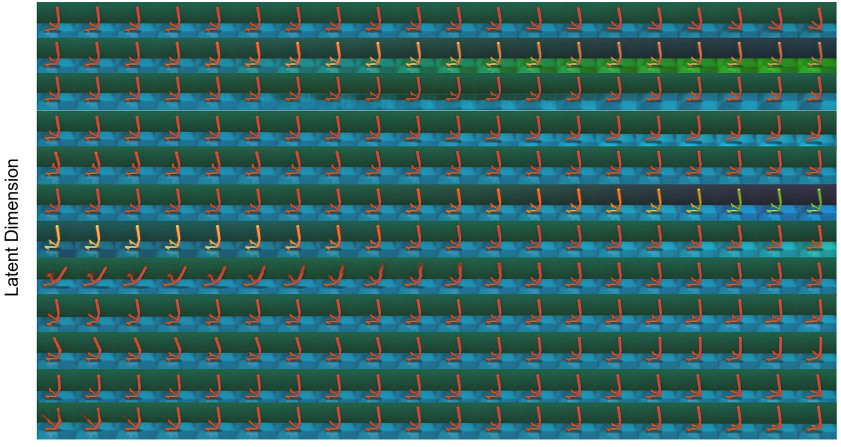

Figure 8: Latent traversals of the disentangled latent vector when training ALDA directly on the "color hard" environment.

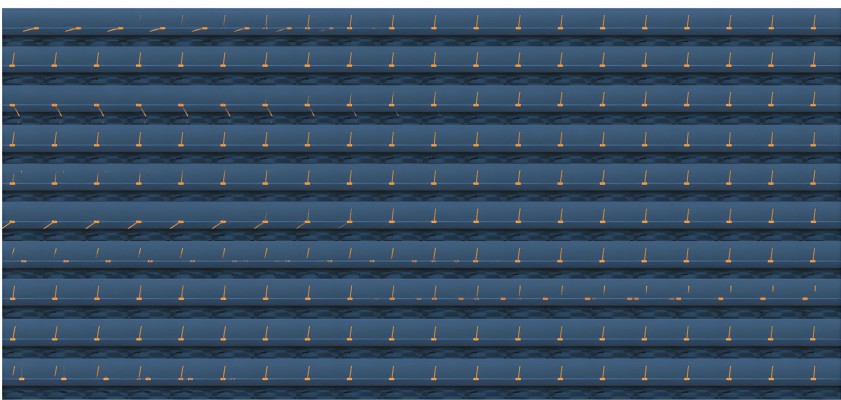

Figure 9: Latent traversals for cartpole balance. Each row corresponds to a latent dimension that is traversed via linear interpolation while all other dimensions are held fixed.

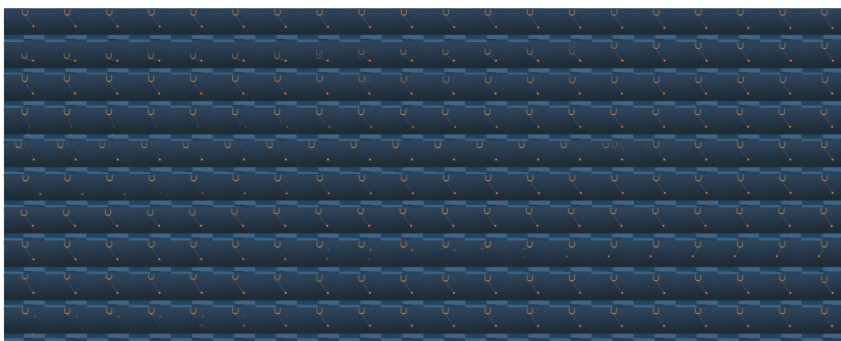

Figure 10: Latent traversals for the finger spin task. Each row corresponds to a latent dimension that is traversed via linear interpolation while all other dimensions are held fixed.

Figure 11: Latent traversals for the ball in cup catch task. Each row corresponds to a latent dimension that is traversed via linear interpolation while all other dimensions are held fixed.

## A.4 BETA STUDY

We perform an analysis of the effects of different $\beta$ values on ALDA's performance. The memory retrieval dynamics are reintroduced here for the reader's reference:

$$z_{d_j} = \text{Softmax}(-\beta L_1(f_\theta(\mathbf{o})_j, \mathbf{v}_j)) \odot \mathbf{v}_j.$$

Small values of $\beta$ result in a more even distribution of the probability mass between latent values per codebook, which implies that the output will be a weighted sum of different latent values (or memories under the Hopfield interpretation). We choose three different values, $\beta = (1, 10, 50)$, and compare with the main result ($\beta = 100$) presented in the paper on the Walker domain, shown in 12. To our surprise, lower $\beta$ values have little to no effect on generalization performance and, in fact, increase training performance. This perhaps challenges the assumptions made in Hsu et al. (2023) about the requirements of disentanglement via latent quantization, but admittedly requires further analysis, which we leave to future work.

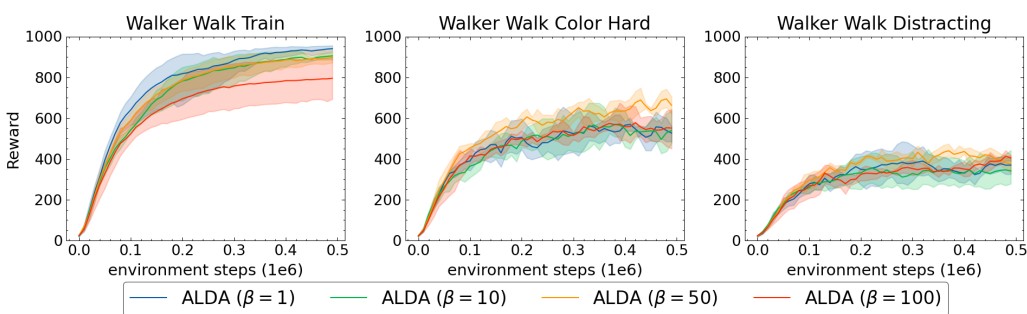

Figure 12: ALDA is trained on the Walker domain with different values of $\beta$, with evaluations periodically performed on the "color hard" and DistractingCS environments. Average of 4 seeds, shaded region represents a 95% bootstrapped confidence interval.

## A.5 FRAMESTACK ABLATION

We provide an ablation comparing against a version of ALDA where the encoder receives as input, and the decoder predicts a stack $k = 3$ of frames, i.e., the observation size is $(9 \times 64 \times 64)$. Since the downstream 1D-CNN layer is no longer necessary, we remove this layer from the variant. We refer to this variant as "ALDA (framestack)" and present results on the Walker domain in Figure 13. We also provide latent traversal visualizations of ALDA (framestack), shown in Figure 14.

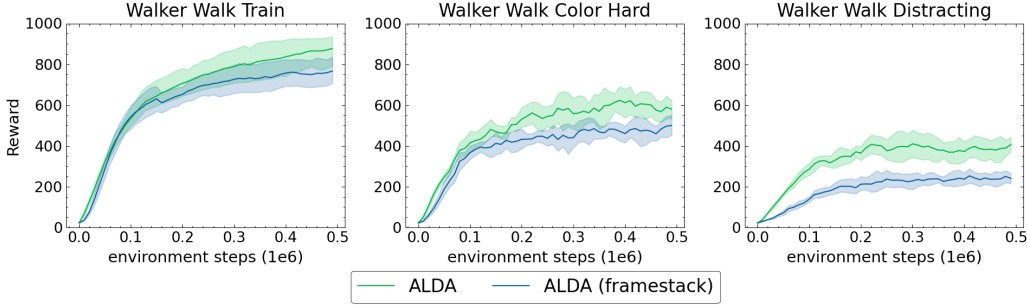

Figure 13: Ablation comparing ALDA to ALDA (framestack) on the Walker domain. Average of 4 seeds, shaded region represents a 95% confidence interval.

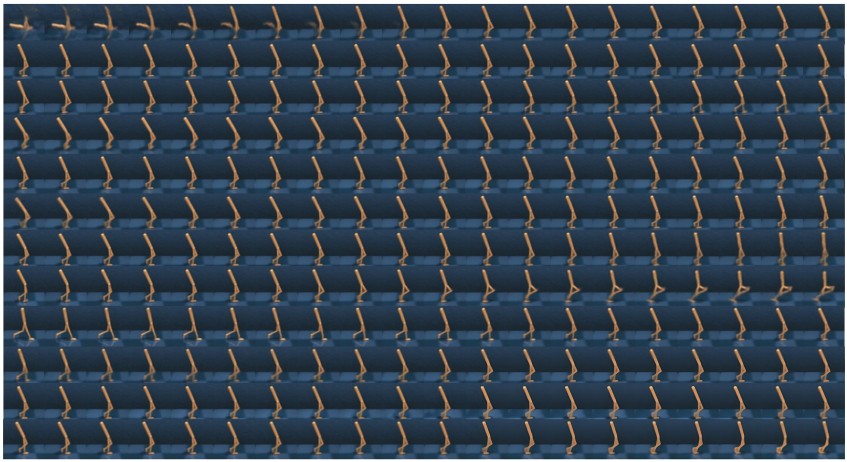

Figure 14: Latent traversal visualizations for ALDA (framestack).

From the plots, we can see that while the generalization performance of ALDA (framestack) does not degrade entirely, it does suffer when compared to ALDA. The latent traversal visualizations show that ALDA (framestack) does not disentangle the dynamics of individual bodies of the agent the way ALDA does. Many latent dimensions, for example, latent dim 8 (5th row from the bottom), affect two or more aspects of the agent when interpolating the latent values. One possible explanation is that ALDA (framestack) does better at capturing and disentangling *temporal* information, given that it sees a stack of consecutive frames, but struggles to disentangle sources of singular images corresponding to non-temporal information, whereas ALDA excels at the latter.

## A.6 HYPERPARAMETERS AND MODEL ARCHITECTURES

Our SAC implementation is based on Yarats & Kostrikov (2020).

### A.6.1 ACTOR AND CRITIC NETWORKS

Following Yarats & Kostrikov (2020), we use double $Q$-networks, each of which is a 3-layer multi-layer perceptron (MLP) with 1024 hidden units per hidden layer and GeLU activations after all except the final layer. The actor network is similarly a 3 layer MLP with 1024 hidden units per layer and GeLU activations on all but the final layer.

### A.6.2 ENCODER, DECODER, AND LATENT MODEL

We use the same encoder/decoder architectures as Hsu et al. (2023), with the exception that we replace all leaky ReLU activations with GeLU. We instantiate the codebooks for the latent model with values evenly spaced between [-1, 1].

### A.6.3 HYPERPAREMETERS

We list a set of common hyperparameters that are used in all domains.

| Parameter | Value |
|---|---|
| Replay buffer capacity | 1e6 |
| Batch size | 128 |
| Latent model temperature $\beta$ | 100 |
| Number of latents $|z_d|$ | 12 |
| Number of values per latent $V_j$ | 12 |
| Encoder weight decay $\lambda_\theta$ | 0.1 |
| Decoder weight decay $\lambda_\phi$ | 0.1 |
| Frame stack | 3 |
| Action repeat | 2 for finger spin otherwise 4 |
| Episode length | 100 |
| Observation space | (9 x 64 x 64) |
| Optimizer | Adam |
| Actor/Critic learning rate | 1e-3 |
| Encoder/Decoder learning rate | 1e-3 |
| Latent model learning rate | 1e-3 |
| Temperature learning rate | 1e-4 |
| Actor update frequency | 2 |
| Critic update frequency | 2 |
| Discount $\gamma$ | 0.99 |

Table 1: Common hyperparameters for SAC and ALDA.

## A.7 ALDA PSEUDOCODE

---

**Algorithm 1** ALDA Forward Pass

---

**Input:** Observation $\mathbf{o}$, encoder $f_\theta$, latent model $l_\psi$, history encoder $h_\gamma$.

$\mathbf{o} \in \mathbb{R}^{B \times Ck \times H \times W} \rightarrow \mathbf{o} \in \mathbb{R}^{Bk \times C \times H \times W}$ // rearrange the framestack dimension
$z_{cont} \in \mathbb{R}^{Bf \times n_z} \leftarrow f_\theta(\mathbf{o})$
$z_d \in \mathbb{R}^{Bf \times n_z} \leftarrow l_\psi(z_{cont})$ // association step using the latent model
$z_d \in \mathbb{R}^{Bf \times n_z} \rightarrow z_d \in \mathbb{R}^{B \times f \times n_z}$ // rearrange the framestack dimension
$z_d \leftarrow h_\gamma(z_d)$ // encode temporal information
**return** $z_d$

---

**Algorithm 2** Associative Latent Dynamics

---

**Input:** Continuous latent vector $z_{cont} \in \mathbb{R}^{Bk \times n_z}$, latent model $l_\psi$

**for** $i \leftarrow 1$ **to** $n_z$ **do**
  $\mathbf{w}_i \leftarrow \mathrm{Softmax}(-\beta L_1(z_{cont_i}, \mathbf{v}_i)) \odot \mathbf{v}_i$ // compute weights for how similar $z_{cont_i}$ is to each value in the $i'th$ codebook.
  $z_{d_i} \leftarrow \sum_{|\mathbf{v}_i|} \mathbf{w}_i$
**end for**
**return** $z_d$

---

Algorithm 1 contains pseudocode for ALDA's forward pass. The observation $\mathbf{o}$ can be an in-distribution sample during training, or an OOD sample during evaluation. Post-training, when presented with in-distribution samples, the association step is unlikely to significantly change $z_{cont}$, since $z_{cont}$ will map very close to the values learned by the latent model $l_\psi$. However, when presented with OOD samples, $z_{cont}$ is more likely to change since certain dimensions of $z_{cont}$ may map far away from the corresponding dimensions of the latent model. Algorithm 2 shows how the latent model $l_\psi$ maps potentially OOD continuous latent vectors to in-distribution values using modern Hopfield retreival mechanisms.

## A.8 LATENT STUDY

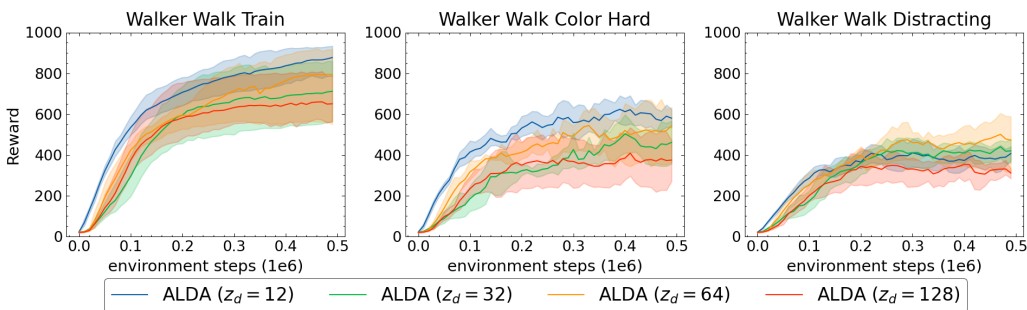

Figure 15: Training (left) and evaluation (middle and right) results of increasing the dimensionality of ALDA's latent space on the Walker Walk task. Results are averaged over 4 seeds. Shaded region represents a 95% bootstrapped confidence interval.

We examine the effects of increasing the dimensionality of the latent space on the performance of the Walker "Walk" task and present the results in Figure 15. The baseline model ($z_d = 12$) performs the best on the training and "Color Hard" evaluation tasks, and that performance drops as the dimensionality of the latent space increases. Since disentangled representation learning methods try to approximate the number of ground truth sources of variation of the data distribution, it is possible that setting $z_d$ to values far away from the number of true sources can cause the performance degradation we observe in Figure 15.

## A.9    CRITIC GRADIENT ABLATION

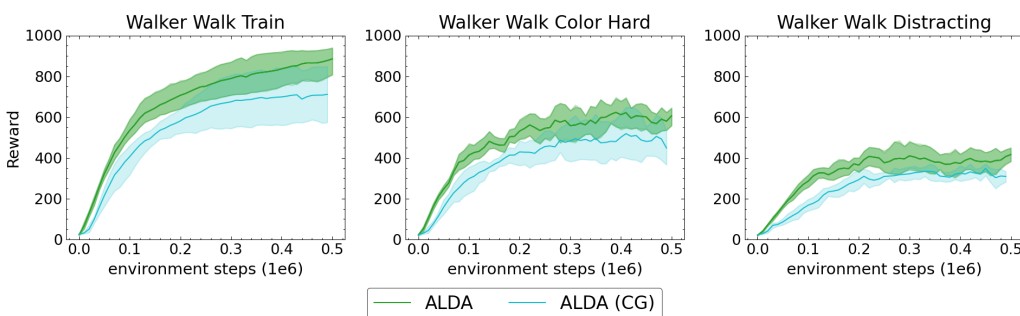

Figure 16: Ablation of backpropogating the critic gradients to the encoder (ALDA CG) compared to the standard model. Results are averaged over 4 seeds. Shaded area represents a 95% bootstrapped confidence interval.

In this experiment, we examine the effects of backpropagating the critic's gradients through the latent model and back to the encoder. The results are presented in Figure 16. We refer to the ALDA variant with critic gradients enabled as "ALDA (CG)" and compare on the Walker "Walk" task. ALDA (CG) performs worse on all training and evaluation tasks. We suspect that backpropagating the critic gradients to the encoder affects the ability for the encoder-decoder pair to disentangle sources of the image distribution, since disentangled representation learning methods typically study disentanglement in (Variational) Autoencoders without competing auxiliary objectives.

## A.10  SVEA + RANDOM CONV AUGMENTATION

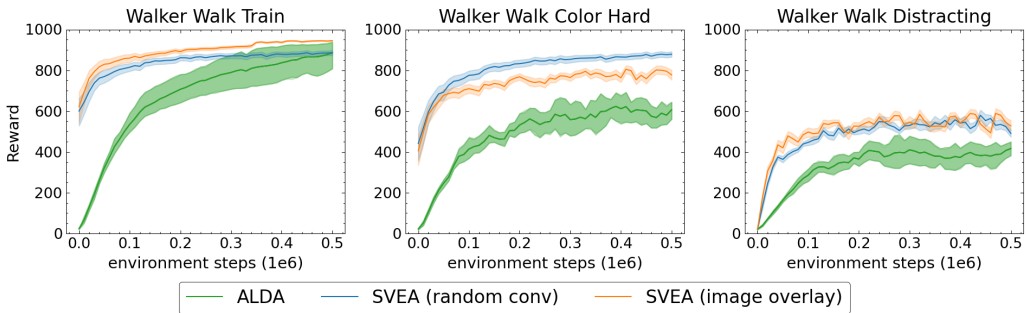

Figure 17: Comparison of ALDA to SVEA using the "rand conv" data augmentation technique. Results are averaged over 4 seeds. Shaded region represents a 95% bootstrapped confidence interval.

We compare ALDA to SVEA using the "random convolution" data augmentation technique, which applies random convolutions to the input observation, changing the colors of the agent and background. SVEA (image overlay) as presented in the main paper is also included as a baseline. The results are presented in Figure 17. We find that SVEA (random conv) performs slightly better on the "color hard" evaluation task compared to SVEA (image overlay), and roughly the same on the "Distracting CS" evaluation task.

