# OpenReview forum: "Zero Shot Generalization of Vision-Based RL Without Data Augmentation"
_ICLR.cc/2025/Conference — Submitted to ICLR 2025_

### Official Review · Reviewer_BAip · 2024-10-27

**Soundness:** 3
**Presentation:** 3
**Contribution:** 3
**Rating:** 6
**Confidence:** 4

**Summary:**

The paper proposes Associative Latent DisentAnglement (ALDA), a novel method for zero-shot generalization in vision-based reinforcement learning (RL) without data augmentation. ALDA combines disentangled latent space representations with an associative memory mechanism inspired by biological systems like the hippocampus. This enables RL agents to generalize to out-of-distribution (OOD) environments by recalling previously seen factors of variation, without need of augmentation in training. ALDA outperforms several state-of-the-art methods in tasks involving distribution shifts, demonstrating zero-shot generalization without using data augmentation.

**Strengths:**

Originality:
The idea of using disentanglement and associative memory for RL to achieve zero-shot generalization is relatively less explored. It is nice formulation to combine both for RL. Unlike existing methods that rely heavily on data augmentation, ALDA achieves generalization without it.

Quality:
The paper is well-researched. It shows data augmentation is one form of disentanglement. It proposed new methods to combine disentanglement and association. The experimentation is carried with ablation studies on different component of the method.

Clarity:
The paper is generally clear, with a well-structured narrative that explains the problem, the motivation, and the solution. It draws inspiration from biological system which is well articulated

Significance:
Its significance lies in that it presents a method for generalization without augmentation. It would be very useful under the case that we cannot anticipate what shift will happen.

**Weaknesses:**

The papers lacks a more thorough discussion of its failure cases.

Suggestion:
1. What if you vary distracting intensity, how would that change the relative results, especially compared to SVEA.
2. I also notice it perform worse than SVEA in the training environment, can you add interpretations on that?

**Questions:**

1. In figure 6, top half, the robot starts from different pose when traversing the latent code. what if it starts from the same as in the bottom half

2. What if we compare the proposed method to data augmentation method that uses simpler augmentation, such as color jitter or random crop? For example

Kostrikov, Ilya, Denis Yarats, and Rob Fergus. "Image augmentation is all you need: Regularizing deep reinforcement learning from pixels." arXiv preprint arXiv:2004.13649 (2020).

Hansen, Nicklas, and Xiaolong Wang. “Generalization in Reinforcement Learning by Soft Data Augmentation.” 2021 IEEE International Conference on Robotics and Automation (ICRA), IEEE, 2021. Crossref, https://doi.org/10.1109/icra48506.2021.9561103.

You can also test of SVEA only using simpler augmentation.

I understand that the main contribution of this paper is generalization without augmentation, it is okay that it is not performing as good.

3. Can you combine the proposed approach with data augmentation? How would you do that?

4. Can you combine disentanglement / association with dynamics? How would do you that?

---

> ### Author Response · Authors · 2024-11-15
> **Response to Reviewer BAip**
>
> **What if you vary distracting intensity, how would that change the relative results, especially compared to SVEA.**
>
> Increasing the distracting intensity increases the degree of camera perturbations. We expect both methods to perform proportionally worse. We are running this experiment and will notify the reviewers when it has been added to the manuscript.
>
> **I also notice it perform worse than SVEA in the training environment, can you add interpretations on that?**
>
> We have several hypotheses for why this might be the case. ALDA’s latent space is discretized to a finite number of bins, whereas we are working with high-dimensional continuous control tasks. This discretization, while necessary for achieving disentangled representations and performing associations at test time, may cause approximation errors. In addition, as mentioned in the discussion section, the disentangled representation literature has primarily focused on using reconstructions in (V)AE’s with other soft constraints to disentangle the latent space, and thus struggles to encode temporal information across images. While we provide an initial solution in the form of a 1D CNN layer to capture temporal information, it is perhaps not as efficient as a method that can both capture and disentangle the temporal information. However, we have yet to design or come across such a method.
>
> **In figure 6, top half, the robot starts from different pose when traversing the latent code. what if it starts from the same as in the bottom half**
>
> The bottom and top figures are different training runs. The initial configuration of the robot does not affect our ability to perform latent traversals.
>
> **What if we compare the proposed method to data augmentation method that uses simpler augmentation, such as color jitter or random crop? For example**
>
>  We are running SVEA with the “random conv” augmentation and will provide the results to the reviewer once they are available.
>
> **Can you combine the proposed approach with data augmentation? How would you do that?**
>
> Yes. Rather than applying color distortions to the data, we train directly on the “color hard” environment and show qualitative results in Figures 6 (bottom) and 8. This effectively simulates training on additional data the same way applying color distortions would. To clarify, we mean data augmentation in the sense that our algorithm will scale with more varied data. However, it is unclear whether certain specific data augmentation techniques, such as randomly overlaying images from a different data distribution onto the training distribution, which results in an unnatural image, is something that ALDA can effectively model. We have updated the manuscript to clarify this point.
>
> **Can you combine disentanglement / association with dynamics? How would do you that?**
>
>
> We assume the reviewer is referring to the agent dynamics. In this case, ALDA is likely doing this with the agent dynamics and not just task-irrelevant variables. The sources of the data distribution for DMControl include agent-dynamics such as rigid-body orientations. We see empirical evidence that ALDA learns to disentangle and gain fine-grained control over individual rigid bodies in Figure 6, Appendix A.2, and Appendix A.3.

---

> > ### Comment · Reviewer_BAip · 2024-11-27
> >
> > Thanks to the authors for replying in the rebuttal. The explanation and additional experiments are helpful in improving the quality of this work. The score is not changed.

---

### Official Review · Reviewer_yGHM · 2024-10-28

**Soundness:** 4
**Presentation:** 2
**Contribution:** 3
**Rating:** 6
**Confidence:** 3

**Summary:**

A common way to improve the generalization ability of an agent in reinforcement learning is through data augmentation. However, collecting data can be costly with increasingly varied tasks.
This paper achieves zero-shot generalization by learning disentangled representations, where each latent feature dimension corresponds to an unique source of variation in the input image.
Inspired by neuroscience, the authors incorporate associative memory into the original representation learning loss in QLAE, and build a novel framework called Associative Latent DisentAnglement (ALDA) for RL.
ALDA is evaluated on various control tasks with two types of variations (color-hard and distracting backgrounds), and compared to state-of-the-art methods.
While its performance lags behind data-augmentation-based methods for generalization, it outperforms previous approaches of learning disentangled representations.

**Strengths:**

- Improving generalization by learning better representations is not a new concept.
For instance, Zhang et al. [2021] aim to encode only task-relevant information from input images.
However, the idea of training disentangled representations specifically for generalization, particularly without data augmentation, is a novel approach.
Recently, numerous studies have explored the effects of scaling laws on enhancing generalization, yet methods for achieving generalization without exposure to out-of-distribution data remain relatively under-explored.

- The idea of considering memory for generalization is intuitive and the author build an interesting connection between the disentangled representation learning framework and associative memory network.

A. Zhang, R. McAllister, R. Calandra, Y. Gal, and S. Levine. Learning invariant representations for reinforcement learning without reconstruction. ICLR 2021

**Weaknesses:**

- According to the paper, the strong disentanglement requires that each dimension in the latent representation corresponds to a unique source of the variation in the image.
As for the weak disentanglement, a more clear and precise definition (line 216 to line 219) might be helpful for the reader to understand.
For example, one natural question to ask is if several dimensions in the latent representation $\textbf{z}$ correspond to one unique source, is this a weak disentanglement?
I would appreciate it if you could provide concrete examples for these two types of disentanglement.
This would help readers better understand the key concepts and their implications for the method.
- And the relationship between disentanglement and generalization ability is not so clear to me.
According to the line 233 to line 235, if I understand correctly, weak disentanglement can also lead to generalization.
Is strong disentanglement a necessary condition for generalization or is weak disentanglement sufficient?
If not, please explain more about the benefits and drawbacks of using strong disentanglement vs. weak disentanglement?
- In the section of experimental results, an explanation is provided for the performance gap between ALDA and SVEA.
However, as mentioned in the paper, the proposed method can definitely be combined with data augmentation.
So, it will strength the paper if the results of combining ALDA and data augmentation are presented.
- minor: missing punctuation on line 305.
In figure 2, $n_s$ is not defined in the paper.
I guess it is $n_z$ on line 290.

**Questions:**

- In Figure 2, the encoder is only updated by the ALDA loss, as indicated by the green arrow and the latent model is updated by both the ALDA loss and the critic loss, as indicated by the red arrow.
Is there a reason for this? Any ablation study for this design choice?
- On line 345, according to "the scalar codebooks Z, which can be interpreted as predetermined memories", are the codebooks trained or pre-defined?
If trainable, are they trianed with equation 8 such that $V$ are actually trained parameters?
If pre-defined, how the values are chosen?
- Is there an intuitive connection between equation 7 and Figure 1?

---

> ### Author Response · Authors · 2024-11-15
> **Response to Reviewer yGHM Part 1**
>
> **According to the paper, the strong disentanglement requires that each dimension in the latent representation corresponds to a unique source of the variation in the image. As for the weak disentanglement, a more clear and precise definition (line 216 to line 219) might be helpful for the reader to understand. For example, one natural question to ask is if several dimensions in the latent representation correspond to one unique source, is this a weak disentanglement? I would appreciate it if you could provide concrete examples for these two types of disentanglement. This would help readers better understand the key concepts and their implications for the method.**
>
> Thank you for bringing this to our attention. We have updated the manuscript (Section 3) to include a more formal definition of weak disentanglement. To address your question, if several latent dimensions each encode only one source, but the source is the same (i.e. the latents are redundant), we would not consider this weak disentanglement. By defining weak and strong disentanglement, we wish to make a distinction between latent spaces where one or more but not all dimensions encode multiple sources i.e. weak disentanglement and latent spaces where each dimension encodes at most one source i.e. strong disentanglement. Consider a 5-dimensional latent space for a dataset containing five sources. In this example, $z=<(\hat{s_0}, \hat{s_1}), (\hat{s_2}), (\hat{s_3}), (\hat{s_3}), (\hat{s_2}, \hat{s_4})>$ would be weakly disentangled because dimensions 1 and 5 encode two sources. $z = <(\hat{s_0}), (\hat{s_2}), (\hat{s_1}), (\hat{s_3}), (\hat{s_4})>$ would be strongly disentangled, because each dimension encodes for only one latent source. We claim that, assuming linear independence between the sources, the latter is ideal for generalization because it achieves the greatest representative power of the data distribution.
>
> **And the relationship between disentanglement and generalization ability is not so clear to me. According to the line 233 to line 235, if I understand correctly, weak disentanglement can also lead to generalization. Is strong disentanglement a necessary condition for generalization or is weak disentanglement sufficient? If not, please explain more about the benefits and drawbacks of using strong disentanglement vs. weak disentanglement?**
>
> In some cases, weak disentanglement is sufficient. Data augmentation methods achieve a type of weak disentanglement that is sufficient for generalization on those specific benchmarks, as demonstrated by the SVEA baseline. These methods remove distractor information and only model task-relevant variables in the latent space, and so there is at least a partial factorization between task-relevant and task-irrelevant sources.. However, there is no guarantee that task-relevant sources i.e. agent dynamics are disentangled. In some cases this is an ok assumption to have, particularly when the task-relevant sources are not linearly independent. However the main drawback as we point out in the paper is that this approach requires an excessive amount of additional data and finetuning for training stability just to achieve weak disentanglement. We simply wish to show that there exists a body of work in the disentangled representation learning literature that achieves strong disentanglement in a much more data efficient way by exploring algorithmic and model changes rather than the brute force approach of collecting more data. Strong disentanglement, definitionally, will achieve the factorization of task-relevant and task-irrelevant variables just by trying to factorize all sources of the data distribution. There are, of course, downsides to this approach, in that it requires more inductive biases in the model architecture, it traditionally requires one to know the sources a priori (although we found a way around this issue through empirical analysis), and it struggles to capture temporal information.
>
> Beyond being robust to distribution shifts, strong disentanglement is thought to be one of the components of compositional generalization. For example, if a human is presented with a red apple, and is able to factorize “color”  from “object”, it is nontrivial to imagine a purple or a black apple despite not being presented with this data (and assuming those colors have been seen in other contexts). This would be extrapolative generalization, whereas traditionally when the deep learning literature refers to generalization, it refers to interpolative generalization. This may be a promising future research direction in other ML fields, such as data efficient generative modeling, world modeling for agents, etc. capable of extrapolative generalization.

---

> > ### Author Response · Authors · 2024-11-15
> > **Response to Reviewer yGHM Part 2**
> >
> > **In the section of experimental results, an explanation is provided for the performance gap between ALDA and SVEA. However, as mentioned in the paper, the proposed method can definitely be combined with data augmentation. So, it will strength the paper if the results of combining ALDA and data augmentation are presented.**
> >
> > Figure 6 (bottom) and Figure 8 (appendix A.3) were intended to be qualitative results showing our method combined with data augmentation. In this case, the random changes in color on environment resets are the additional training data, or “data augmentations”, provided to ALDA, which learns to model color information separately from other sources of the data distribution. To clarify, in this context we refer to data augmentations as providing additional training data to the algorithm. We are not sure how ALDA will perform with specific augmentation techniques such as random overlays of images from a separate data distribution, since this technique produces unnatural image training data with scene discontinuities. We have clarified this in the updated manuscript.
> >
> > **minor: missing punctuation on line 305. In figure 2, is not defined in the paper. I guess it is on line 290.**
> >
> > We have fixed the punctuation error on line 305. The n_s variable in figure 2 is defined in section 2.2.
> >
> > **In Figure 2, the encoder is only updated by the ALDA loss, as indicated by the green arrow and the latent model is updated by both the ALDA loss and the critic loss, as indicated by the red arrow. Is there a reason for this? Any ablation study for this design choice?**
> >
> > Most disentangled representation learning methods train a (V)AE with reconstruction loss combined with strong regularization to disentangle the latent space. We were uncertain how backpropagating the gradients of the critic would affect the ability of the disentanglement component of ALDA to effectively disentangle the sources, and thus decided to prevent the gradients from flowing to the encoder. We will provide an ablation for this design choice and add it to the manuscript as soon as the results are available.
> >
> > **On line 345, according to "the scalar codebooks Z, which can be interpreted as predetermined memories", are the codebooks trained or pre-defined? If trainable, are they trianed with equation 8 such that are actually trained parameters? If pre-defined, how the values are chosen?**
> >
> > The codebooks are initialized with values evenly spaced between [-1, 1] into $|V_j|$ bins. The parameters are optimizable, but with our specific formulation and the omission of the quantization loss term, they are not updated and remain fixed throughout training.
> >
> > **Is there an intuitive connection between equation 7 and Figure 1?**
> >
> > Yes, the blue arrows are the associative latent dynamics presented in equation 7. Each latent dimension $v_j$ is mapped to the codebook representing the original range of valid latent values (or memories under the Hopfield interpretation) independent of other latent dimensions using the Hopfield dynamics laid out in equation 7.

---

> > > ### Comment · Reviewer_yGHM · 2024-11-25
> > >
> > > I appreciate you time and effort on the response. Considering the novelty of this paper and the improved presentation after revision, I raise my score.

---

### Official Review · Reviewer_BwQf · 2024-11-02

**Soundness:** 3
**Presentation:** 3
**Contribution:** 3
**Rating:** 6
**Confidence:** 4

**Summary:**

This work introduces Associative Latent Disentanglement (ALDA), a zero-shot generalization strategy for vision-based RL free of data augmentation. Inspiring by neurobiology, ALDA enables agents to generalize to out-of-distribution (OOD) activities by combining associative memory with disentangled latent representations. The results of the experiments show that ALDA is just as good at generalization as or better than current augmentation-based methods, especially in difficult OOD settings.

**Strengths:**

1. The paper takes ideas from neuroscience and combines latent disentanglement with associative memory to present a new, augmentation-free method for zero-shot generalization in RL.

2. The methodology is sound, with experiments across challenging tasks that effectively support the primary claims of the paper.

3. The paper is generally clear and well organized, though some technical sections could be further clarified for readability.

4. This work has potential impact by presenting an efficient alternative to data augmentation, likely inspiring further research on generalization in RL without large datasets.

**Weaknesses:**

1. More proof of ALDA's generalization ability would be obtained by extending the studies to more varied settings, such as Procgen.
Procgen's randomly produced levels and diverse game environments present distinct challenges, unlike those found in the DMControl suite. These settings may demonstrate the efficacy of ALDA's generalization across diverse activities, offering a comprehensive assessment of its adaptation to novel and intricate situations.

2. More illustrations or pseudocode could help to clarify the function of associative memory in processing OOD samples. Particularly, pseudocode illustrating ALDA's processing of an OOD sample—from initial encoding to final output—would effectively elucidate the function of associative memory in the generalization process.

3. An assessment of ALDA's effectiveness in real-world settings would demonstrate its applicability and scalability. For example, testing on a robotic manipulation job requiring the robot to generalize to objects of diverse forms, sizes, or textures not seen during training, could effectively showcase its generalization capabilities in practical applications.

**Questions:**

1. In order to show that ALDA is generalizable to a wider variety of OOD contexts, could you test it on other benchmarks, such as Procgen?

2. Could you elaborate on the associative memory dynamics in ALDA, perhaps using a pseudocode or flow diagram, to show how it specifically supports zero-shot generalization?

3. Is the method applicable to an offline reinforcement learning setting, given that online reinforcement learning can be costly and risky in real-world environments? It would be beneficial to address the adjustments necessary for adapting ALDA for offline reinforcement learning, along with the potential obstacles the authors expect in this adaptation.

---

> ### Author Response · Authors · 2024-11-15
> **Response to Reviewer BwQf**
>
> **More proof of ALDA's generalization ability would be obtained by extending the studies to more varied settings, such as Procgen. Procgen's randomly produced levels and diverse game environments present distinct challenges, unlike those found in the DMControl suite. These settings may demonstrate the efficacy of ALDA's generalization across diverse activities, offering a comprehensive assessment of its adaptation to novel and intricate situations.**
>
> We agree that extending the studies to more settings such as Procgen would help validate the capabilities of ALDA. However, the recent literature on zero-shot generalization in vision-based RL, which has formed the majority of our baselines, don’t test on these environments, instead opting for DistractingCS and / or the color shift environments on DMControl only. Finetuning each baseline and producing results on other benchmarks perhaps warrants a broader investigation on the state of zero-shot generalization in RL given current methods, but is out of scope for this paper.
>
> **More illustrations or pseudocode could help to clarify the function of associative memory in processing OOD samples. Particularly, pseudocode illustrating ALDA's processing of an OOD sample—from initial encoding to final output—would effectively elucidate the function of associative memory in the generalization process.**
>
> Thank you for bringing this to our attention. The processing of an OOD sample is exactly the same as the processing of a training sample. The only difference is that the continuous output of the encoder will be different with an OOD sample. When the associative latent dynamics maps the encoder output to the latent codes, the average “distance traversed” for each latent code for OOD samples will be higher than in-distribution samples. We have added pseudocode for both ALDA’s forward pass and the associative latent dynamics in the appendix (A.7).
>
> **An assessment of ALDA's effectiveness in real-world settings would demonstrate its applicability and scalability. For example, testing on a robotic manipulation job requiring the robot to generalize to objects of diverse forms, sizes, or textures not seen during training, could effectively showcase its generalization capabilities in practical applications.**
>
> We agree with this completely! However, testing our method in real environments requires other considerations, such as finetuning our method to work with RGB(D) data only and not proprioceptive state (whereas most, if not all sim2real papers use both), since including proprioceptive state as input somewhat defeats the purpose of our method, and including baselines focused more on sim2real rather than simulated RL environments. In addition, we are considering implementing our approach within other RL frameworks such as on-policy RL to test the generalizability of our approach from an algorithmic perspective. We have plans to carry out this work, but we believe this warrants a separate paper.
>
> **In order to show that ALDA is generalizable to a wider variety of OOD contexts, could you test it on other benchmarks, such as Procgen?**
>
> Please see our response to the reviewer’s comment above.
>
> **Could you elaborate on the associative memory dynamics in ALDA, perhaps using a pseudocode or flow diagram, to show how it specifically supports zero-shot generalization?**
>
> We have uploaded pseudocode showing ALDA’s forward pass and the memory retrieval mechanism accompanied by a short explanation in the appendix, section A.7.
>
> **Is the method applicable to an offline reinforcement learning setting, given that online reinforcement learning can be costly and risky in real-world environments? It would be beneficial to address the adjustments necessary for adapting ALDA for offline reinforcement learning, along with the potential obstacles the authors expect in this adaptation.**
>
> We believe ALDA can be adapted to the offline RL setting. Note that the disentangled representation learning literature typically trains models on “offline” pre-curated datasets, which is a very similar setting to the offline RL problem. One potential challenge is estimating the number of sources (i.e. the size of the latent representation) on large offline-RL datasets that are sufficiently diverse. Ideally ALDA would continuously disentangle new data into encodings of new sources as new data appeared, in which case a fixed vectorized representation would not be sufficient. Other self-supervised disentanglement techniques that bias the model to learn generic object-centric representations without labels or masks such as slot attention [1], combined with some form of associative memory, may be more suitable for the offline-RL setting.
>
> [1] Locatello, Francesco, et al. "Object-centric learning with slot attention." Advances in neural information processing systems 33 (2020): 11525-11538.

---

> > ### Comment · Reviewer_BwQf · 2024-11-21
> > **Response to authors**
> >
> > Thank you for your efforts in addressing the points I raised regarding the paper. I have carefully reviewed the authors' responses and appreciate the time and thought they put into their feedback. However, while I value their efforts, their answers do not change my perspective on the work. As a result, I will be maintaining my original scores.

---

### Official Review · Reviewer_sfzs · 2024-11-03

**Soundness:** 3
**Presentation:** 3
**Contribution:** 2
**Rating:** 5
**Confidence:** 4

**Summary:**

The paper proposes an algorithm based on SAC, that performs latent disentanglement representation and combines it with an associative memory model to achieve zero-shot generalization in vision-based RL. The paper shows that the proposed approach improves upon several baselines when tested in two evaluation environments (color hard and distracting cs). In addition, the authors theoretically prove that data augmentation methods for vision-based RL implicitly perform "weak" disentanglement, and therefore lead to a noticeable zero-shot generalization.

**Strengths:**

* The paper tackles the important challenge of generalization in vision-based RL.

* The paper is well-written and easy to follow.

* The background section clearly explains the different methods for disentanglement representation and associative memory. Also, it provides motivation for using these methods for zero-shot generalization.

* There is a good discussion about the relevant work of zero-shot generalization in vision-based RL.

* The paper visually demonstrates (Figure 6) that the resulting disentanglement representation successfully encodes different data attributes.

**Weaknesses:**

* The method was tested in a narrow set of test environments (color hard and distracting cs). The paper could be improved if the authors would show the benefit of the proposed approach on standard (and more challenging) procedurally generated vision-based datasets for zero-shot generalization in RL, such as Procgen [1] or Crafter [2].

* In lines 180-184, the authors mention that one drawback of previous approaches is that the latent representation excludes all information not relevant to the training tasks, which can hinder adaptation to new tasks that involve information that was previously considered irrelevant. The paper would benefit from a concrete experiment that showcases this failure and demonstrates how the proposed approach handles it.

* Some baselines are missing from the experiment section. For example, a comparison to methods that use the information bottleneck to exclude irrelevant information from the state representation in vision-based RL, such as [3].

* An ablation study of |zd| (the dimension of the latent space) is missing.

[1] Cobbe K, Hesse C, Hilton J, Schulman J. Leveraging procedural generation to benchmark reinforcement learning. In International conference on machine learning 2020 Nov 21 (pp. 2048-2056). PMLR.

[2] Hafner D. Benchmarking the spectrum of agent capabilities. arXiv preprint arXiv:2109.06780. 2021 Sep 14.

[3]  Maximilian Igl, Kamil Ciosek, Yingzhen Li, Sebastian Tschiatschek, Cheng Zhang, Sam Devlin, and 492 Katja Hofmann. Generalization in reinforcement learning with selective noise injection and information 493 bottleneck. In Advances in Neural Information Processing Systems 32, pages 13956–13968, 2019.

**Questions:**

1. Please address the aforementioned weaknesses.

2. Is it possible to utilize an RNN unit instead of frame stacking to incorporate temporal information into the latent code?

3. Section A.2 and Figure 7 are not clear to me. Would you please explain the experimental setting there and the results?

4. How was the hyperparameter search done for all the baselines?

---

> ### Author Response · Authors · 2024-11-15
> **Response to Reviewer sfzs**
>
> **The method was tested in a narrow set of test environments (color hard and distracting cs). The paper could be improved if the authors would show the benefit of the proposed approach on standard (and more challenging) procedurally generated vision-based datasets for zero-shot generalization in RL, such as Procgen [1] or Crafter [2].**
>
> We agree that it would be interesting to see how our method performs on other vision-based datasets curated for zero-shot generalization. However, much of the recent influential work on addressing zero-shot generalization, which we included as baselines, do not report results on these environments, and have focused solely on DistractingCS and/or color shift environments in DMControl. While we would eventually like to expand this work to other domains such as Procgen and improve on the existing algorithm, finetuning our method and all baselines to other domains is an extensive undertaking and out of scope for this paper.
>
> **In lines 180-184, the authors mention that one drawback of previous approaches is that the latent representation excludes all information not relevant to the training tasks, which can hinder adaptation to new tasks that involve information that was previously considered irrelevant. The paper would benefit from a concrete experiment that showcases this failure and demonstrates how the proposed approach handles it.**
>
> It is difficult to show this effect on the DMControl tasks given the lack of scene complexity. Other reviewers have pointed out that showing the performance of this method on real environments with varying objects, colors, lighting conditions, etc. might help showcase ALDA’s capabilities, which we agree with and have planned as future work. We believe real environments would be the more appropriate choice to showcase this effect, for example training to pick and place a single object with various other objects in the background, but then evaluating the model’s ability to perform pick and place on the objects that were in the background.
>
> **Some baselines are missing from the experiment section. For example, a comparison to methods that use the information bottleneck to exclude irrelevant information from the state representation in vision-based RL, such as [3].**
>
> The “RePo” baseline reported in the paper uses an information bottleneck in the form of a KL penalty to exclude irrelevant information from the latent space. We will add a short explanation of the RePo baseline to make this clear and refer the reviewer to [1] for more details on the implementation.
>
> **An ablation study of $|z_d|$ (the dimension of the latent space) is missing.**
>
> Thank you for bringing this to our attention. We will add this ablation to the paper and notify the reviewers when the manuscript has been updated with the additional experiments
>
> **Is it possible to utilize an RNN unit instead of frame stacking to incorporate temporal information into the latent code?**
>
> It should be possible to replace our 1D CNN layer with an RNN, but we didn’t try this since ALDA and several baselines only use 3 consecutive frames, thus not requiring learning of any long-horizon dependencies.
>
> **Section A.2 and Figure 7 are not clear to me. Would you please explain the experimental setting there and the results?**
>
> DMControl provides access to low-dimensional proprioceptive state information as the input representation as opposed to RGB images. Since ALDA attempts to learn a strongly disentangled latent representation, we were curious if any of the dimensions of ALDA’s learned representation from RGB data corresponded to any of the proprioceptive state variables. After training the agent, we performed a single rollout and logged the values of several select latent dimensions over time (top row) and compared them to several select rigid-body orientations over time (bottom row) that came from the ground truth proprioceptive state vector. While we can’t make any strong conclusions given that the encoder can map one-to-many, we do notice oscillatory behavior in these latent dimensions, same as the rigid-body orientations over time. This suggests, at least in a qualitative way, that the latent space is not only disentangling but possibly modeling individual rigid bodies of the agent as if we had used proprioceptive state information to begin with. We think this might be a very fruitful future line of research attempting to recover what has typically been hand-engineered proprioceptive state information (which is inherently disentangled) directly from high-dimensional data.
>
> **How was the hyperparameter search done for all the baselines?**
>
> We used the default hyperparameters provided by the codebases of each respective baseline.
>
>
> [1] Zhu, Chuning, et al. "Repo: Resilient model-based reinforcement learning by regularizing posterior predictability." Advances in Neural Information Processing Systems 36 (2023): 32445-32467.

---

### Author Response · Authors · 2024-11-15
**Message to All Reviewers**

Thank you everyone for providing in depth feedback and questions on the paper. Several reviewers have requested additional experiments. We are currently in the process of running the following experiments and will notify the reviewers when the results have been added to the updated manuscript:

- Ablation study of $|z_d|$
- Ablation on backpropagating critic gradients to the encoder
- Increasing the distracting intensity on Distracting CS and comparing to SVEA
- Using a weaker data augmentation technique such as random-conv with SVEA and comparing to ALDA

In addition, several reviewers have asked for clarity on the associative memory latent dynamics. We have provided pseudocode in the appendix (Section A.7) for both ALDA's forward pass and the associative memory dynamics in the latest version of the manuscript for the reviewer's reference. In the meantime, we will respond to reviewer feedback individually.

---

### Author Response · Authors · 2024-11-25
**Message to All Reviewers (2)**

We have uploaded a new version of the manuscript that contains the requested study of $|z_d|$ and an ablation on backpropagating the critic gradients to the encoder. We are still working on the additional SVEA comparisons and plan to have this before the discussion period ends. In the meantime, we kindly request the reviewers to respond to our comments and examine the existing additional requested experiments and pseudocode. Thank you!

---

### Author Response · Authors · 2024-11-26
**Message to All Reviewers (3)**

Hello all,

We have uploaded results comparing ALDA to SVEA + random convolution data augmentation technique in the appendix (Section A.10).
Unfortunately, the last requested experiment on training SVEA with greater distracting intensity on the Distracting CS environment diverged during training and we will not be able to get these results before the rebuttal period ends. However, we plan to finish these experiments and add them to the final, camera ready version of the paper. To summarize our changes, as per reviewer requests, we have added the following:

- A study of scaling $|z_d|$  (A.8)
- Pseudocode of ALDA's forward pass and associative latent dynamics (A.7)
- Ablation of back propagating the critic gradients to the encoder (A.9)
- Comparison of ALDA to SVEA + "random conv" data augmentation technique (A.10)

Thank you to the reviewers who have already engaged in discussion and responded to our rebuttal. As a reminder, we kindly request all reviewers to examine the additional experiments and figures and respond to our follow-ups before the discussion period ends.

---

### Meta-Review · Area_Chair_31QU · 2024-12-21

**Metareview:**

This paper proposes Associative Latent DisentAnglement (ALDA), a method for zero-shot generalization in vision-based reinforcement learning (RL) without relying on data augmentation. ALDA combines disentangled latent representations with an associative memory mechanism, drawing inspiration from neuroscience to enable agents to generalize to out-of-distribution environments.

The strengths of this paper include addressing the important challenge of zero-shot generalization in vision-based RL and proposing a novel combination of disentangled representations and associative memory. The main weaknesses of this paper include limited evaluation, demonstrations, and analysis.

This paper received mixed scores. During the post-rebuttal AC-reviewer discussion, Reviewer sfzs suggested that additional validations of generalization, including even simple real-world demonstrations, are essential. Other reviewers did not express disagreement, and nobody explicitly championed the paper. Therefore, the decision is to reject this submission.

**Additional Comments On Reviewer Discussion:**

Reviewer yGHM changed the score from 5 to 6 due to the improved presentation after revision and the paper's novelty. Other reviewers did not change their scores.

---

### Decision · Program_Chairs · 2025-01-22

Reject